# Individual, Familial, and Socio-Environmental Risk Factors of Gang Membership in a Community Sample of Adolescents in Southern Italy

**DOI:** 10.3390/ijerph17238791

**Published:** 2020-11-26

**Authors:** Dario Bacchini, Mirella Dragone, Concetta Esposito, Gaetana Affuso

**Affiliations:** 1Department of Humanistic Studies, University of Naples “Federico II”, 80133 Napoli, Italy; mirella.dragone@unina.it (M.D.); concetta.esposito3@unina.it (C.E.); 2Department of Psychology, University of Campania “Luigi Vanvitelli”, 81100 Caserta, Italy; gaetana.affuso@unicampania.it

**Keywords:** antisocial behaviors, community violence witnessing, parental rejection, self-serving cognitive distortions, youth gang membership

## Abstract

Despite the growing social alarm generated by the recurrent news concerning violent episodes involving youth gangs, systematic research in Italy in this field, especially within a psychological framework, is still limited. Following a social-ecological approach, the present study aimed at investigating the role of self-serving cognitive distortions (CDs), parental rejection, and community violence witnessing in youth gang membership (YGM). Furthermore, we examined the mediating and/or moderating role of YGM in the association between risk factors and involvement in antisocial behaviors (ASBs). A community sample of 817 adolescents attending middle and high schools in a high-risk urban area in Southern Italy (46.9% males; 53% middle school students; M_age_ = 14.67; SD = 1.65) were involved in the study. One hundred and fifty-seven participants (19.2%) were found to be gang members. Employing counterfactual-based mediation analysis, we found that CDs and community violence witnessing were directly associated with YGM and ASBs. The association between CDs and ASBs was mediated by YGM. Parental rejection was directly related to ASBs but not to YGM. A significant interaction effect between parental rejection and YGM was found, revealing that high levels of parental rejection, along with being a gang member, amplified the involvement in ASBs. These findings pointed out that distorted moral cognitions and the experience of violence witnessing within the community may represent a fertile ground for gang involvement. Both individual and contextual factors should be considered in order to implement interventions aimed to prevent adolescents’ risk of joining a gang.

## 1. Introduction

Youth gang membership (YGM) is a serious public health concern in many countries around the world. In recent years, the phenomenon has received increasing prominence in the media, highlighting the detrimental consequences of YGM for adolescent development and for the safety of the social community. Despite the increasing attention among criminologists and researchers to YGM, a clear theoretical framework of the phenomenon is far from being reached, and there is a frequent claim in the literature for a re-conceptualization of the construct (e.g., [1,2]). A long series of questioning issues marks the debate on YGM, ranging from the definition itself [3], the causal or incidental relationship with delinquent behavior [4], and the competing models to explain dynamics and motivations underlying joining a gang (see [2]).

Prevalence data concerning YGM reported by Departments of Justice are impressive. In 2010, 1,059,000 youth gang members were estimated in the US [5], corresponding to 2% of the young population, with a peak of 5% at the age of 14 years. YGM is also an alarming phenomenon in Europe, even though the lack of a shared definition of youth gang affiliation and the lack of a common database of crimes committed by youth deviant gangs does not allow one to fully estimate prevalence data. The most reliable source comes from a large survey carried out in 30 European countries involving over 40,000 adolescents; 4.4% of the European community sample were estimated as members of deviant youth groups [6]. Gang membership can be considered a transient phenomenon, with a turnover rate of 36% of youth that annually join and exit from the gangs [5,7], but many individuals persist actively in gangs till 30 years of age [8], and joining a gang during adolescence can be considered a stepping stone for a future delinquent career [9].

The literature identified the main risk factors pushing an adolescent to become a gang member, usually coded in five domains: individual, family, peer, school, and community [10]. Nevertheless, each of these domains comprises a series of sub-domains whose specific influence on gang membership, even tested across research, produced inconsistent results. Moreover, YGM is a phenomenon embedded within a socio-culturally specific background, and, consequently, behaviors and motivations are highly variable across cultural sites. Since most research has been performed in the US, where the phenomenon is deeply rooted in the socio-cultural background, it is not obvious that the US gangs’ features would overlap with the gangs’ features in other countries. According to the summative research report of the EU Gangs program [11], Italian youth gang association is quite different from the representation of American gangs. In Italy, there are very few groups that have some, or often none, of these features. For this reason, in order to avoid any confusion, many European researchers prefer to use terms like “troublesome youth group” [12,13] or “delinquent or deviant youth group” (e.g., [6,14]) instead of the word “gang”.

In the present research, we aim to contribute to the gang literature by examining YGM in the Italian context, as little research has been carried out on this topic, with even less adopting a psychological perspective. Specifically, we aimed at systematically investigating the role of individual, family, and neighborhood/community risk factors for YGM, examining socio-psychological constructs such as self-serving cognitive distortions (CDs), parental acceptance–rejection, and community violence exposure. These constructs have received limited attention in previous studies. Nonetheless, this limited existing research has produced inconsistent results. Furthermore, we examined the role of YGM in mediating and/or moderating the association between risk factors and the involvement in antisocial behaviors (ASBs), thus contributing to the debate around the still open question of whether YGM enhances ASBs or whether only previously antisocial individuals deliberately choose to become gang members.

### 1.1. Youth Gangs in Italy

In recent years, a great deal of emphasis has been given in Italy, especially in Southern Italy, to crimes and violence committed by youth gangs.

However, despite the social alarm, very few systematic studies have been conducted to investigate the gang phenomenon in Italy. Such studies have often been limited in using an ethnographic approach with specific subgroups of gangs, such as soccer supporters [15] or immigrants [16]. A more systematic contribution comes from the research by Blaya and Gatti [17]. In a study performed with a large community sample of over 5000 adolescents, the authors found that 17% of participants self-evaluated belonging to a gang. Furthermore, they found that about one third of gang members were females and that group affiliation was associated with high involvement in different kinds of illegal acts. Moreover, a multicentric study [6] was carried out to estimate the rate of belonging to a “deviant youth group,” defined according to the Eurogang criteria as “any durable, street-oriented youth group whose own identity includes involvement in illegal activity” [18]. In Italy, 4.4% of participants satisfied the criteria to be defined as members of a deviant youth group involved in delinquency, violence, and alcohol or drug use. This percentage was similar to the mean percentage across countries.

### 1.2. Moral Cognitions and Youth Gang Membership

The assumption that individuals committing deviant behaviors tend to justify themselves from the responsibility and the consequences of their acts has a long tradition in the criminological science since Sykes and Matza’s [19] theory of “neutralization.” Individuals who act in an antisocial way try to overcome the discrepancy between their behavior and internalized social norms and beliefs by cognitive rationalization processes that deny or minimize the seriousness of their acts or justify them in some way [20]. This assumption has been conceptualized in various ways. According to the social-information processing model [21], aggressive individuals justify their behaviors by attributing hostile intentions to others. Mechanisms of moral disengagement were identified by Bandura [22,23] as self-regulating mechanisms through which individuals disengage their own moral principles in order to legitimate their immoral or aggressive behavior. Gibbs [24] proposed the model of self-serving CDs defined as “inaccurate or biased ways of attending to or conferring meaning upon experiences” [25] (p. 1). More specifically, Barriga and Gibbs [26] distinguished between primary and secondary self-serving CDs. Primary (i.e., self-centered) distortions serve as main motivators of aggressive behaviors because they are characterized by an egocentric bias that reflects more immature moral judgment stages stemming from self-centered attitudes, thoughts, and beliefs. Secondary distortions, labeled as blaming others, minimizing/mislabeling, and assuming the worst, take the form of pre- or post-rationalizations serving to cognitively overcome dissonance between individual moral standards and behavioral transgressions and neutralizing potential empathy and guilt, thus avoiding damage to one’s self-image and facilitating deviant behaviors.

The recourse to neutralization techniques in order to misattribute the evaluation of deviant behaviors has also been described in relation to gang affiliation (e.g., [27,28]), examining, in particular, the role of moral disengagement mechanisms. A seminal study by Esbensen and Weerman [27] conducted with adolescents from the US and the Netherlands found that gang members scored significantly higher on moral disengagement items than non-gang youth in both of those countries. Moreover, in a study conducted with a community sample of the UK high school students, Alleyne and Wood [29] found that specific moral disengagement mechanisms, such as euphemistic labeling, displacement of responsibility, and blaming others, were more frequently used by gang members compared to their non-gang counterparts. Nevertheless, in a subsequent study, Alleyne and Wood [30] found that moral disengagement mechanisms did not directly influence gang-related crime, but their influence was fully mediated by anti-authority attitudes. On the other hand, the mechanism of dehumanization of the victim mediated the relationship between gang affiliation and crime, suggesting that being a gang member amplifies the ingroup–outgroup conflicts justifying crime [31]. In another study, also conducted in UK, Niebieszczanski et al. [32] found a significant difference between gang and non-gang members in all the eight moral disengagement mechanisms. Although the above mentioned findings were not entirely consistent, empirical evidence points out that gang members were more likely to utilize moral disengagement strategies, thus suggesting that the status of a street gang member may act as “fertile ground” for the disengagement of moral self-sanctions [2].

Collectively, these findings evidenced the facilitating role of moral neutralization mechanisms in YGM but, to our knowledge, no study was conducted adopting the theoretical framework of self-serving CDs, as conceptualized by Gibbs’s model [24], suggesting the interest in exploring this new field of research.

### 1.3. Parental Acceptance–Rejection and Youth Gang Membership

A great number of studies focused on the role of family factors in influencing YGM. Nevertheless, the view that family contributes to gang membership has not gained universal acceptance [11]. In this regard, in the paradigmatically titled paper “A question of family? Youth and gangs”, Young, Fitzgibbon, and Silverstone [33] argue that connecting the family to gang membership leads to far from conclusive results, since the etiology of gang formation and criminality cannot simply be reduced to poor home environments or “broken” families. The aim of the present paper is not to focus on the wide range of family risk factors that the literature has examined in relation to YGM, such as family structure and socio-economic disadvantage [34], living with both biological or adoptive parents [35], lack of parental discipline [4], weak parental control, and lack of supervision of the child’s activities [18,36,37].

In the current research, we focus on the parent-child relationship, specifically on the adolescents’ perceived rejection by their parents as a factor promoting joining a gang. A number of studies performed within the theoretical framework of Rohner’s Parental Acceptance–Rejection Theory (PARTheory; [38]), recently renamed Interpersonal Acceptance–Rejection Theory (IPARTheory; [39]), assume that high perceived parental acceptance and high perceived parental rejection are associated with positive and negative developmental outcomes, respectively (e.g., [40,41]). Some studies carried out in Italy have confirmed the association between parental rejection and externalizing behaviors [42,43]).

Moving on to the specific literature on youth gangs, a recurrent statement is that adolescents rejected by their parents are attracted to gangs because they would find in the gang a sort of surrogate family [44], which could satisfy their emotional needs [45] and provide the emotional support that is not available from the family of origin [46]. Moreover, the sense of belongingness given by gang affiliation would satisfy the need for social connections and survival [47].

Despite the fascinating claim of this statement, research findings seem inconsistent. For instance, Thornberry [48] found that low parent–adolescent attachment (here considered as a proxy of parental acceptance) and poor parental supervision lead to gang involvement, and Baskin, Quintana, and Slaten [49] found that family belongingness showed a bivariate correlation with gang membership. Compared to non-gang youth, gang members were found to be significantly more likely to live in families characterized by lower levels of parental warmth and inconsistent discipline [50]. By contrast, other researchers found that the security of attachment to parents was unrelated to later gang membership [51] or lost its influence when accounting for other family variables [35,52,53].

Walker-Barnes and Mason [54], in a longitudinal study with three subsamples of white, black, and Hispanic American adolescents, evidenced a significant impact, even if weak, of parental rejection on YGM, with a moderating effect of ethnicity on this association. More specifically, parental rejection was related to gang involvement in black adolescents but not in the other ethnic group, suggesting the need to consider the role of cultural variables in explaining family effects. Furthermore, in a subsequent study [55], parental warmth was found to moderate the association between gang involvement and two forms of maladaptive outcomes (i.e., minor delinquency and substance use), although in the opposite direction respect to that hypothesized by the authors because high levels of parental warmth were found to increase the strength of the relationship between gang involvement and adolescent problem behavior. A speculative/suggestive explanation provided by the authors was that higher levels of parental warmth, under these circumstances, may be perceived by gang-involved youths as approval of their behavior and inadvertently reinforce children’s maladaptive behavior.

### 1.4. Neighborhood Violence Exposure and Youth Gang Membership

Research has posited that YGM aligns within disadvantaged and socially disorganized communities [4,13,56,57]. Although a growing amount of research has demonstrated the association between community violence exposure and delinquent behavior (for a review, see [58]) over and above other contexts of violence exposure [59], few studies have examined the specific association between community violence exposure and joining a gang. Young people growing up in violent neighborhoods may develop a form of “pathologic adaptation” to violence [60] due to the recourse to moral neutralization mechanisms in order to neutralize the negative feelings resulting from the impact of violent experience. In addition, community violence exposure represents a risk factor for delinquency for many other reasons. For instance, according to social learning theory [61], chronic exposure to violent models makes youth more likely to learn patterns of aggressive behavior perceived as adaptive, and, at the same time, chronic community violence exposure negatively affects self-regulative capabilities over time [62].

A few studies examining the direct association between community violence exposure and gang membership found that gang belongingness was associated with a higher fear of being victimized in the neighborhood [63], increasing the likelihood to experience social problems [50]. The association between the perception of unsafety in the neighborhood and gang involvement was systematically examined by Merrin, Hong, and Espelage [64] in a study with a large sample of over 17,000 middle and high school students in the US. Using a socio-ecological approach and a multilevel technique of analysis, perceived neighborhood unsafety was an independent predictor of gang involvement, accounting for other individual, school, family, and peer factors.

It must be noted that it might be difficult to disentangle the effect of community violence exposure from neighborhood social disorganization factors, because youths who frequently witness violence typically reside in urban and inner-city neighborhoods where they experience economic and psychosocial problems that increase the risk of joining gang activities [29,65,66].

Finally, the relationship between community violence and gang affiliation seems complex, since adolescents often report joining a gang to receive a sense of affiliation and support as well as protection and financial gain [67], which are lacking in their dangerous neighborhood. At the same time, joining a gang increases the likelihood of being exposed to neighborhood violence [68,69] and, therefore, perpetuates the cycle of violence [70].

### 1.5. Youth Gang Membership as a Risk Factor for Involvement in Antisocial Behaviors

Different models explain the relationship between gang affiliation and delinquent behaviors, which is well summarized in Thornberry et al.’s [4] distinction among a selection model where gangs select and recruit members who are already delinquent [36,71], a facilitation model where gangs provide opportunities for delinquency to youth who were not delinquent beforehand [72,73], and an enhancement model where gang members are recruited from a population of high-risk youth who, as gang members, become more delinquent [72,73]. Given the variety of methodological and measurement approaches, it is not easy to compare studies that have generated inconsistent results. However, most studies seem to agree that certain youth may be predisposed to gang membership but once in the gang increase their involvement in delinquency, thus supporting the enhancement model [74].

### 1.6. The Present Study

Despite the growing social alarm generated in Italy by the recurrent news concerning violent episodes involving youth gangs, systematic research in this field, especially within a psychological framework, is still limited. The aim of the present study is to fill this gap by investigating YGM in a community sample of adolescents from Southern Italy. We contribute to the literature on juvenile gangs by investigating the role of YGM in mediating and/or moderating the relationship between individual (i.e., self-serving CDs), family (i.e., parental rejection), and neighborhood/community (i.e., exposure to community violence as a witness) risk factors and ASBs.

Based on the theoretical considerations discussed above, the following main research questions were addressed in this study: (i) “Are individual self-serving CDs, perceived parental rejection, and witnessing community violence related to YGM?” We hypothesize a unique contribution in determining YGM from self-serving CDs aimed at neutralizing inhibitory processes of deviant behaviors; negative feelings of being rejected by parents enhancing the need of seeking belonging and protection through gang affiliation; and community violence exposure leading adolescents to joining a gang and to adhesion to street culture in order to seek protection in the dangerous neighborhood where they live; (ii) “Is YGM related to ASBs when accounting for other antisocial-related risk factors?” According to the enhancement model, we hypothesize that YGM could mediate the association between risk factors and ASBs. Namely, we expect a direct association of risk factors with YGM that, in turn, affected ASBs. In addition, we also test a moderating role of YGB in the association between risk factors and ASBs.

Furthermore, as previous studies have found gender- and age-based differences in CDs, parental rejection, violence exposure, and ASBs as well as in joining a gang, we controlled the results for potential gender and school grade effects. Specifically, prior studies have consistently found that males and older youth are at greater risk for community violence exposure than females and younger children (e.g., [75]); males typically self-report more CDs and ASBs [76] and are at greater risk for parental rejection as well as for joining a gang [77] compared to females. However, although we expect more gang involvement in males than in females, in recent years female involvement in gangs has generated great attention [78], and an increasing number of studies suggest that females are becoming more involved in gangs than in the past [14].

## 2. Materials and Methods

### 2.1. Participants

The study design involved eighth and eleventh graders of two middle and three high schools in Arzano, a relatively small town located in the metropolitan area of Naples (Italy). This area is characterized by serious social problems such as high unemployment, school dropout, and the presence of organized crime [79]. Overall, episodes of crime and violence such as robberies, threats, extortions, presence of criminal organizations, tracking, and drug possession are causing growing social alarm in the Italian context. According to the most recent official data from the Public Security Department of the Italian Ministry of the Interior (2018), the number of reported crimes in Italy is approximately 6600 per day. The metropolitan area of Naples is the second in Italy for reported crimes, with 56,891 for every 100,000 inhabitants.

The sample for the current study consisted of 817 adolescents, 383 males and 434 females (46.9% males; 53% middle school students), assessed in 2015. The age of participants ranged from 12 to 18, with a mean age of 14.67 (SD = 1.65). The socio-economic characteristics of the sample were representative of the Southern Italian population. Specifically, approximately 66% of participants’ fathers and mothers had a low level of education (i.e., middle school diploma or less), 25% had a high school diploma, and about 9% had a university degree. Moreover, 43% were unemployed or did a low-profile job.

### 2.2. Procedure

Approval of the University Institutional Review Board (IRB; file reference number 3/2020, dated 13 January 2020) was obtained for collecting data, which took place during spring 2015 after receiving parents’ written consent and adolescents’ assent, in accordance with the ethical principles of the Italian Association of Psychology (AIP). The questionnaires were administered in the classroom, during ordinary class sessions, by trained assistants and took approximately 30 min to complete. To reassure participants about reporting sensitive information and to encourage honest reporting, a complete guarantee of confidentiality was emphasized. Additionally, participants were informed about the voluntary nature of participation and their right to discontinue at any point without penalty.

### 2.3. Measures

#### 2.3.1. Gang-Related Measures

##### Gang Membership

Gang membership was measured by asking respondents, “Have you ever been or are you now a member of a gang?” (1 = Yes, 2 = No). The use of self-nomination as an indicator of gang membership was utilized in previous studies on gang membership [35,80] and was demonstrated to be a robust measure capable of distinguishing gang from non-gang youth [3,81]. In addition, as we were aware of the concerns related to definitional issues [3], we followed the Eurogang definition including the involvement in illegal activity as a requisite for gang membership. We used more restrictive criteria by asking participants to respond to two items related to the frequency of gang-involved assaults (i.e., “damage or destroy property” and “get in fights with other gangs”) (see [7]). An affirmative response to one of these two items, in addition to the affirmative response to the question related to gang affiliation, led to designation as an “organized gang” member.

#### 2.3.2. Gang-Related Risk Factors

##### Self-Serving Cognitive Distortions (CDs)

Participants were asked to respond to the 39 items of the How I Think Questionnaire (HIT; [25]; Italian validation by [82]), measuring self-serving CDs. For each item, participants were asked to indicate their agreement on a six-point Likert scale (from 1 = disagree strongly, to 6 = agree strongly). Sample items were “People need to be roughed up once in a while” and “Everybody breaks the law; it’s no big deal.” An overall HIT score was computed by averaging the 39 item scores, with a higher score indicating higher levels of self-serving CDs. Cronbach’s alpha was 0.97.

##### Perceived Parental Acceptance–Rejection

Perceived parental acceptance-rejection was measured with the child–adolescents’ version of the Parental Acceptance–Rejection Questionnaire (PARQ–Child; [83]). Participants responded to two versions of the PARQ to provide their assessments of maternal (PARQ–Mother) and paternal (PARQ–Father) acceptance-rejection. The PARQ is a 24-item self-report instrument to which participants were asked to indicate, using a four-point scale (from 1 = never, to 4 = every day), the frequency of perceived mother and father parenting behaviors in terms of warmth–affection, hostility–aggression, neglect–indifference, and undifferentiated rejection (e.g., “My [mother/father] makes me feel wanted and needed”; “My [mother/father] goes out of [her/his] way to hurt my feelings”). All subscales showed acceptable alpha levels, both in the mother’s (Cronbach’s alpha ranged from 0.77 to 0.86) and in the father’s (Cronbach’s alpha ranged from 0.76 to 0.87) version of the questionnaire. Given the high correlations between each subscale of maternal acceptance–rejection and the paternal counterparts (Pearson’s r ranged from 0.69 to 0.80), we computed an overall score of parental rejection by averaging the eight mother and father warmth–affection (reversed), six hostility–aggression, six neglect–indifference, and four undifferentiated rejection item scores. A higher score indicated greater perceived parental rejection. Cronbach’s alpha was 0.85 for the total scale.

##### Community Violence Exposure

Exposure to community violence as a witness was self-reported using the Exposure to Community Violence Questionnaire (ECVQ; [62]), a scale assessing community violence exposure through witnessing. Items were originally selected from a review of the Community Experience Questionnaire by Schwartz and Proctor [84] based on their relevance to the specific urban context and adapted and validated in Italian research [62]. Adolescents were asked to report violent incidents that had occurred during the last year and only serious real-life events from their neighborhoods and their communities, not incidents from movies or television or from day-to-day conflicts with other peers at school. The ECVQ included six items to which adolescents were asked to report, using a five-point scale (from 1 = never, to 5 = more than five times), the frequency of their witnessing violence in the neighborhood during that time period. A sample item for witnessing community violence was, “How many times have you seen somebody get robbed?” Cronbach’s alpha was 0.88.

#### 2.3.3. Gang-Related Outcome

##### Antisocial Behaviors (ASBs)

To detect ASBs, we used the Antisocial Behavior Questionnaire (ASBQ; [59]) made up of 18 items to which the subject was asked how often she/he had engaged in a series of ASBs. All the items referred to behaviors in which an adolescent can be personally involved (e.g., “gambling,” “substance use,” “involvement in fights,” “bullying peers,” “not paying for the ticket on public transit”). The subjects were asked to respond to each item using a five-point Likert scale, ranging from 1 (never) to 5 (very often) based on the frequency with which they had engaged in that behavior during the last year.

An overall ASBs score was computed by averaging the 18 item scores, with a higher score indicating more adolescent engagement in ASBs. The factor structure of the scale was assessed with confirmatory factor analysis. The single factor model had an acceptable fit (YBχ2(135) = 396.553, *p* < 0.001; CFI = 0.93; TLI = 0.92; RMSEA = 0.05; SRMR = 0.04, 90% C.I. [0.04, 0.05]). As in previous studies [59], the instrument showed good reliability indices, with a Cronbach’s alpha of 0.94.

#### 2.3.4. Control Variables

Sociodemographic characteristics of the sample were collected by asking participants to indicate their own age, gender (1 = male, 2 = female) and school grade (1 = middle, 2 = high school students).

### 2.4. Data Analysis

Before testing our hypotheses, we firstly identified within the sample participants who self-nominated as gang members based on an affirmative response to the question “Have you ever been or are you now a member of a gang?.” Subsequently, according to the Eurogang definition [18], we used a more restrictive criteria of gang membership, detecting among those who self-nominated gang members those subjects (i.e., “deviant gang members”) who affirmed that the gang to which they belonged had been involved “at least once” in illegal activities (i.e., “damage or destroy property” and “get in fights with other gangs”).

The univariate normality of data distribution was tested, finding that no study’s variables approached skewness > |3| or kurtosis > |10|. To compare means of a continuous variable (i.e., ASBs) in three independent comparison groups (deviant gang members vs. non-deviant gang members vs. non-gang members), a preliminary one-way Univariate Analysis of Variance (ANOVA) through Bonferroni′s multiple-comparison test was performed. The eta-squared statistic was used to establish effect size. Levels of eta-squared (Ƞ^2^) effect size were interpreted as follows: small (0.01), medium (0.06), and large (0.14) effect size. A *p*-value probability level of <0.05 was adopted throughout.

Furthermore, we compared gang vs. non-gang members by gender and school grade using a set of Chi-square statistics in IBM SPSS 21 (IBM, Armonk, NY, USA) which allows to test hypotheses about distributions of categorical data. Associations among the study’s variables were performed through Pearson correlations correcting threshold levels of significance using Bonferroni correction for multiple comparisons. The mediation and moderation analysis were carried out in Mplus 8 [85]. Questions about mediation and interactions were addressed using the counterfactual framework for causal inference [86,87]. Counterfactual methods have been used in the previous gang literature, given their strength in the approximation of an experimental design [74]. This approach has been described as particularly challenging in that it allows to make causal inferences taking into account outcomes (generally referred to as “potential outcomes”) that would have occurred under conditions that are different from those actually observed in empirical data. Furthermore, the counterfactual approach provides a versatile framework to derive direct and indirect effects when the causal relationships under study involve binary mediators, as in our case.

In the counterfactual perspective, the total effect on the outcome can be decomposed into three parts: pure direct effect (PDE), pure indirect effect (PIE), and mediated interaction. With a continuous independent variable, as in our case, this approach requires to choose two levels of this variable for the estimation of effects. For the current study’s purposes, independent variables were centered around the mean, and the mean value and the value one standard deviation above the mean were chosen for the estimation of effects. Based on this approach, the estimation of the PDE compares what would happen to the outcome when the independent variable changes from the mean to one standard deviation above the mean, setting, for each individual, the mediator to whatever it would be for that individual in the condition of no change in the independent variable. The estimation of the PIE expresses how much the outcome would change on average if the independent variable were fixed at one standard deviation above the mean, but the mediator changed from the level it would take if the independent variable would be set at the mean value to the level it would take if the independent variable would be set at one standard deviation above the mean. In both cases, the term “pure” refers to the fact that these effects are disentangled from the effect of the interaction between the independent variable and the mediator, which is accounted for the mediated interaction. More specifically, the mediated interaction accounts for the reciprocal part where the independent variables affect the mediator, but the effect of the mediator on the outcome is amplified by values that the independent variables assume.

The study’s hypotheses were tested in two steps: First, we investigated the effects of CDs, parental rejection, and witnessing community violence on YGM and ASBs, as well as the effect of YGM on ASBs. Then, we examined pure direct (unexplained by YGM) and indirect effects (via YGM) based on the counterfactual approach. As a second step, we considered the interaction between the independent variables and YGM in the model, disentangling pure mediation effects from mediated interaction effects. Adolescent school grade and gender were used as confounding variables in all the models.

## 3. Results

### 3.1. Preliminary and Descriptive Analysis

Among 325 self-nominated gang members (39.8%; M = 156, M_age_ = 14.65 years, SD = 1.67), only 157 subjects (i.e., “deviant gang members”; 19.2%; M = 96, M_age_ = 14.60 years, SD = 1.72) reported that their gang had been involved “at least once” in illegal activities.

A comparison between deviant gang members, non-deviant gang members, and non-gang members on engagement in ASBs was performed. The findings of one-way ANOVA are reported in Table 1. Overall, a medium effect size, as indicated by the eta-squared index (Ƞ^2^  = 0.06), was found. Some participants who self-nominated as gang members could be more appropriately equated to those not belonging to a gang, since, more probably, they intended the gang as synonymous with a network of friends who usually meet and spend a lot of time in several activities, without any particular antisocial connotation.

Therefore, in subsequent analysis, we used as a key gang-related construct a dummy variable including two groups: non-gang members (n = 660; M = 287, M_age_ = 14.69 years, SD = 1.63), including those who did not self-nominate as gang members and those who self-nominated as members of a gang never involved in illegal activities, and gang members, whose involvement in illegal activities is part of their group identity.

In order to examine gender- and age-related differences between the two groups (non-gang members vs. gang members), a set of Chi-square statistics was performed, revealing that significant differences by gender (ꭓ^2^ (1) = 15.89; *p* < 0.001) and school grade (ꭓ^2^ (1) = 4.40; *p* < 0.05) emerged, with 96 males out of 383 (25.07%) becoming gang members compared to 61 females out of 434 (14.06%), and middle school participants were more involved in a gang than high school participants (95 vs. 62; 21.94% vs. 16.15%, respectively).

Finally, descriptive statistics and Pearson correlations to investigate associations among the study’s variables were carried out. As can be observed in Table 2, after Bonferroni correction for multiple comparisons, it was found that individual (i.e., self-serving CDs) and contextual (i.e., parental acceptance-rejection and community violence witnessing) risk factors were positively intercorrelated with each other and with ASBs; YGM was positively interrelated with ASBs and with both self-serving CDs and community violence witnessing whereas any association emerged between parental acceptance–rejection and YGM. Furthermore, gender (coded 1 = male, 2 = female) was negatively associated with self-serving CDs, community violence witnessing, and ASBs while school grade (coded 1 = middle, 2 = high school students) was positively associated with self-serving CDs, parental acceptance–rejection, and ASBs, with males and high school students reporting significantly higher levels on all considered variables. Finally, consistent with previous findings from Chi-square statistics, YGM was negatively associated with gender, with males being more likely to belong to the gang.

### 3.2. Individual and Contextual Risk Factors of Youth Gang Membership

Results from the logistic regression estimating the effects of individual (i.e., self-serving CDs) and contextual (i.e., parental acceptance–rejection and witnessing community violence) risk factors of YGM are reported in Figure 1. As can be observed, self-serving CDs and witnessing community violence had a significant effect on increasing the probability of being involved in a gang. Parental rejection had no significant association with YGM. The linear regression model estimating the effect of YGM on ASBs showed a significant effect, with gang members showing higher levels of engagement in ASBs.

### 3.3. Counterfactual Effects: Pure Direct and Indirect Effects

Counterfactual analyses from the study indicated strong evidence for direct effects, and only marginally supported mediated effects. The PDEs were significant for CDs (estimate = 0.17; 95% CI = 0.12 to 0.22, *p* < 0.001), parental rejection (estimate = 0.03; 95% CI = 0.02 to 0.04, *p* < 0.001), and witnessing community violence (estimate = 0.09 95% CI = 0.06 to 0.12, *p* < 0.001). With respect to indirect effects, only the PIE of CDs was significant (estimate = 0.03; 95% CI = 0.01 to 0.05, *p* < 0.05), indicating that YGM partially explained why CDs led to ASBs.

Allowing for interactions between the independent variables and mediator suggested that no interactions significantly affected mediated effects (all *ps* > 0.05). We found only one significant interaction effect between parental rejection and YGM (estimate = 0.33; 95% CI = 0.08 to 0.58, *p* < 0.001). The Johnson–Neyman technique (J-N; [88]) was used to probe for interaction and to identify ranges of values of the moderator (two SDs above and below the mean were used) for which the interaction effect was significant. As depicted in Figure 2, the J-N procedure highlighted two regions of significant moderations, revealing that the higher the parental rejection, the more being in a gang was associated with ASBs. On the other hand, for those who reported very low levels of parental rejection, being in a gang predicted having lower than average ASBs scores.

## 4. Discussion

Youth gang membership is a widespread phenomenon all over the world. Youth gangs are often identified with US gangs, but gangs’ features, motivating factors, and behavioral correlates vary across socio-cultural contexts. A few studies have investigated YGM in Italy, although media and Departments of Justice frequently raise the alarm due to an emerging phenomenon in Italian society, especially in the South of the country, where the highly rooted presence of organized crime (e.g., ‘Mafia’, ‘Camorra’, in slang) represents a fertile background promoting the shift from spontaneous youth affiliation in a gang to affiliation in dangerous and highly structured criminal organizations.

In the present study, we fill a gap in the literature investigating YGM in an Italian community sample, the differential role of a set of risk factors in predicting YGM, and the complex interplay between YGM and risk factors on involvement in antisocial behaviors. Despite the increasing concern for crimes and violence committed by youth gangs in the Southern Italy, there is a lack of systematic research. Alarming episodes range from bullying and vandalism to more serious physical assaults against peers and criminal activities such as extortion and selling drugs. Great prominence has been given by the media to a recent phenomenon nicknamed “stesa” in slang (“layer”, as a proxy for translation), indicating a group of young people on motorbikes who suddenly appear in the street running and shooting gunshots into the air. Italian media label these groups “baby gangs” to emphasize the young age of the actors. YGM has specific features in the South of Italy, where organized crime is deeply rooted within the society and many youths are fascinated by the models of the adult criminal gangs. This phenomenon has been well illustrated by the novelist and journalist Roberto Saviano [89], who in his romance *La paranza dei bambini* (*The piranhas* in the published English version) describes the escalation of a group of friends aged 12–14 years in a wild gang controlling criminal activities in their territory, even contrasting older gangs.

### 4.1. Rates, Features, and Definitional Issues

The present study was conducted with a community sample of eighth and eleventh grade students attending public schools in the metropolitan area of Naples in Southern Italy. Thirty-six percent of participants responded “yes” to the question “Have you ever been or are you now a member of a gang?” but only 19.5% were considered to belong to juvenile gangs. We classified gang members only as individuals who affirmed that their gang was involved in fights against other gangs and/or in illegal activities. The high discrepancy between the two percentages confirms the ambiguity of the term “gang” outside the US context [18]. The term “gang” is commonly used by non-Anglo-American native speakers, but it is often intended as synonymous with “group of friends”. For this reason, we agree with researchers who warn against misuse of the term; a single gang-related item should be used with caution to avoid false positive classification. The rates of affiliation in our sample overlap with a study carried out in Italy [17] but are higher than another study [6] adopting more restrictive criteria to define a gang.

Moreover, as regards the socio-demographic characteristics of youth gangs, our sample revealed that males were more involved in gangs than females. This finding is common in the literature, although, beyond the statistical differences, it is noteworthy that 14% of females were involved (vs. 25% of males), suggesting that the role of females in gangs is not marginal and requires increasing attention by researchers [78]. Younger people were more involved in gangs than older people. This finding is consistent with the literature positing 14 years of age as the critical age when the rates of YGM reach their peak [5]. One possible reason might be that the onset of risk behaviors seems to anticipate over the years, maybe due to corresponding anticipation of puberty onset which is characterized by an imbalance between the hyper-responsive reward processing regions, that make younger adolescents more easily aroused and inclined towards sensation seeking, and slowly maturing cognitive control regions, which triggers risky behavior, such as joining a youth gang [90].

### 4.2. Risk Factors of Youth Gang Membership

Following a social-ecological approach [91,92,93] according to which individual and contextual factors, as well as their interaction, predict human behavior, we hypothesized in the present study the independent contribution of self-serving CDs (i.e., individual factor), parental acceptance–rejection (i.e., familial factor), and community violence exposure (i.e., environmental factor) on YGM. Our hypotheses were partially confirmed. Self-serving CDs and community violence exposure were independently associated with joining a gang, whereas parental acceptance–rejection was not.

Findings concerning the role of self-serving CDs are consistent with previous studies [28,29,31,32] carried out in the US and the UK, showing the role of mechanisms of self-justification and neutralization of deviant behaviors (i.e., moral disengagement in the aforementioned studies) in enhancing joining a gang. However, it must be noted that the cross-sectional nature of our study, as well as of those previously mentioned, does not fully allow to determine whether the recourse to non-moral cognitions precedes gang affiliation or whether being in a gang contributes to develop self-serving cognitive strategies of neutralization of deviant behaviors. It is reasonable to hypothesize that pre-existing cognitive mechanisms, such as the tendency to minimize the consequences of deviant behavior as well as the tendency to perceive the world as a hostile, threatening, or dangerous place, find fertile ground in gang affiliation. The gang world exacerbates the ingroup–outgroup conflict according to which “others” are viewed as hostile and threatening; consequently, the feelings of guilt when attacking other groups are minimized in the name of their gang’s identity, honor, and safety. Therefore, the construct of self-serving CDs can be a novel contribution in the understanding of individual factors associated with joining a gang.

Furthermore, in the present study there is also evidence that being exposed to community violence is independently associated with gang membership, accounting for the other individual and familial risk factors. This result is in line with previous studies [30,69] finding a direct association between neighborhood unsafety and the likelihood to become a gang member. Our findings add a novel contribution to the gang literature focusing on a specific component of the general concept of neighborhood disorganization, supporting the idea that living and being exposed to community violence are intrinsically linked to gang membership. This link can be explained by several theoretical models. According to the social learning theory [61], witnessing violence promotes the interiorization of models of behavior, emphasizing the violence as a successful way to manage human interrelationship. Similarly, the “pathologic adaptation model” [60] posits that repeated exposure to community violence leads to normalization of violence through mechanisms of moral neutralization, which, in turn, facilitate engagement in future episodes of violence. In light of the social-ecological approach [92], the environment plays a crucial role in interaction with individual characteristics in determining human behavior. More specifically, the role of the social capital and the moral capital—milestones of the social ecological model [93]—should be considered. High levels of neighborhood violence undermine the human resources addressed to the investment of personal and collective resources toward justice/virtue and the sense of belongingness and protection of individuals within the community.

Regarding family influence on YGM, we underlined in the introduction the inconsistent results from the literature. It is still a questionable issue whether youth join a gang driven by the need to find a support that they do not have in the family of origin. In our sample, this assumption was not corroborated. A negative family climate did not seem to give a significant contribution to YGM. More specifically, in our sample, perceived feelings of rejection by parents (mother and father) did not result associated to YGM, whereas parental rejection, as we will discuss in the following section, was positively associated with involvement in ASBs. Although our results are consistent with other studies [35,53], they should be generalized with caution because the family role could be highly context related. In other words, the statement that adolescents rejected by their parents are attracted to gangs because they would find in the gang a sort of surrogate family [44] could be true in some contexts but not in others. Further studies are needed to disentangle the apparent contraddiction in our results, since parental rejection is associated to ASBs but not to YGM. Maybe a relevant role could be played by the high value that family bonds have in the Italian culture, which might not be easily replaced by bonds developed within a youth gang.

### 4.3. Youth Gang Membership and Involvement in Antisocial Behaviors

Antisocial behaviors were directly associated with both risk factors (i.e., individual, family, and community) and YGM. These findings fully support our expectations and are consistent with a large body of literature in which adolescents involved in ASBs show higher levels of CDs [26,94], as well as community violence exposure [95], and perceived parental rejection than their peers not involved. At the same time, adolescents in gangs exhibit more ASBs than their peers not involved.

Our hypothesis that YGM mediated the association between risk factors and ASBs was partially confirmed. Gang membership demonstrated partial mediation between CDs and ASBs. In other words, CDs directly affect ASBs but at the same time promote the affiliation to a gang, increasing the risk of ASBs. Although not conclusive, this result seems to support the enhancement model [73] in which gang members share similar risk factors of antisocial individuals who are not gang members, but, at the same time, being a gang member increases the likelihood to commit antisocial acts.

Finally, the results concerning the role of family rejection are more complex because, although associated with ASBs, it does not seem to support the choice to joining the gang. However, we found a moderating effect of gang membership on the association between parental rejection and ASBs. Namely, when the levels of parental rejection are higher, being a gang member amplifies the likelihood to be involved in ASBs, whereas for lower levels of parental rejection, even gang members are less likely to be involved in ASBs. This finding suggests the importance of considering the interaction between risk factors and gang membership (in line with the enhancement model). However, further studies are needed in order to better understand the role of the family in the relationship between YGM and ASBs.

### 4.4. Strengths, Limitations, and Future Directions

The present study has a number of strengths and limitations. This is the first study, to our knowledge, to have systematically investigated the YGM phenomenon in Southern Italy, where YGM is causing high social alarm. Given that most studies on YGM concern the US context, it is important to expand research throughout multiple geographical and socio-cultural areas to investigate the characteristics of juvenile gangs. The recent constitution of the Eurogang network within the Euopean Society of Criminology is a promising perspective in order to compare findings from different socio-cultural areas. The second strength of this study is the concurrent examination of individual and contextual risk factors, which have been extensively investigated in relation to youth deviance but have received little empirical attention referring to YGM in previous studies. Lastly, we contributed to the debate concerning the causal or incidental role of gang affiliation in ASBs, accounting for other antisocial-related risk factors.

Nevertheless, our research has a series of limitations to note. First, the research relies exclusively on self-report measures that may be subject to social desirability biases, with adolescents as the only informants. Indeed, referring to the tendency to make self-serving CDs and to involvement in ASBs, as well as in youth gangs, it is known (e.g., [96]) that adolescents are more careful about their social image than other age groups, and may be unlikely to report behavior that displays them in a negative light. Moreover, regarding the parental acceptance–rejection experiences and exposure to community violence, a more objective and comprehensive assessment of rejection and violence in the everyday lives of adolescents might be required. For instance, official data from national census agencies and police departments’ reports might be collected, with specific regard to neighborhood levels of crime. Future studies should allow to interpret more accurately the findings utilizing a multi-informant approach (e.g., parents’ reports) jointly with self-report measures.

A second important limitation concerns the cross-sectional nature of the study, which does not allow to clearly determine the direction of the relations between YGM, ASBs, and individual, familial, and socio-environmental variables. There is a great need to collect longitudinal data to determine the causal relationships among variables, clarifying, for instance, whether the high rates of community violence exposure precede YGM or are a consequence. Moreover, longitudinal studies might allow one to investigate which factors are related to persistent vs. transient gang membership. Therefore, future research should take into consideration the developmental trajectories of the constructs investigated to improve explanatory models of gang membership and set up effective prevention programs.

Finally, our sample includes middle and high school students from the same geographical area. We believe that gang characteristics are highly embedded within a specific cultural background, thus more research is needed to confirm the validity of our findings in culturally different populations. Because in the Southern Italian context there is a high rate of school dropout, which is a concurrent factor of YGM, the school might have led to an under-representation of gang members, as previous research suggests that gang youth are indeed likely to be truant [97]. Future research should also include youth who have dropped out of school.

## 5. Conclusions

The findings of the current study extended the literature on juvenile gangs, highlighting the role of individual moral cognitions characterized by a negative view of the world and others and by a systematic misattribution of the seriousness of deviant acts and exposure to violence in the community, which would seem fertile ground for juvenile deviance, on the likelihood of being affiliated with a gang. Moreover, our study provides further empirical support that ASBs are widespread in adolescence, and, consistently with the Eurogang criteria, affiliation with a youth gang promotes ASBs, encouraging the emergence of a “deviant group identity” [18].

We are aware that our study explains only in a small part the phenomenon of YGM. According to the social-ecological model, youth are involved in a multitude of microsystems that could influence their behavior. The role of school experiences, peer relationships and, more in general, the dominant culture within the community where they live should be acknowledged. Moreover, the complexity of the human-environment transaction implied in the genesis of YGM suggests the utility of a multidisciplinary approach, considering the different levels of analysis of the phenomenon (psychological, social, legal, criminological).

Our study provides relevant suggestions for policymakers, school administrators, law enforcement and juvenile justice professionals, mental health specialists, and parents, whose prevention and intervention efforts should be focused on multiple ecological levels, including both individual and contextual factors. In this respect, as CDs and exposure to community violence have been found to play a crucial role in youth gang affiliation in adolescents, our study points to the benefit of school- or community-based approaches that target the strengthening of adolescents’ moral cognition. The Equipping Youth to Help One Another (EQUIP) program [98] is an example of an effective multi-component cognitive-behavioral program developed within Gibbs’s theoretical framework and aimed at educating behaviorally at-risk youth in thinking and acting responsibly by reducing their thinking errors or CDs. Based on positive peer culture, in which individuals feel responsible for each other and help one another, EQUIP is expected to have a great public impact, given that it promotes, in the long term, the development of a non-violent and law-abiding culture, which is a crucial condition for ensuring success in preventing and reducing adolescents’ risk of joining a gang.

## Figures and Tables

**Figure 1 ijerph-17-08791-f001:**
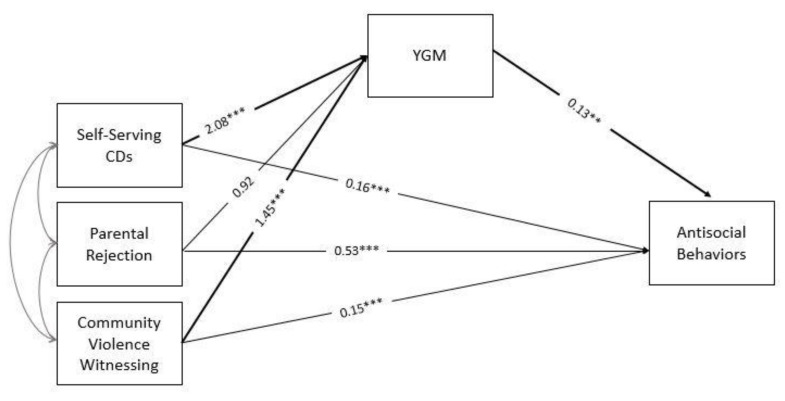
CDs = Cognitive distortions; YGM = Youth gang membership. YGM was coded 0 for non-gang members, 1 for gang members. Control variables are gender and school grade. Reported coefficients refer to odds ratio (for paths linking self-serving CDs, parental rejection and community violence witnessing to YGM) and unstandardized estimates (for the path linking YGM to antisocial behaviors). For the sake of simplicity, relations with control variables are omitted. ** *p* < 0.01, *** *p* < 0.001.

**Figure 2 ijerph-17-08791-f002:**
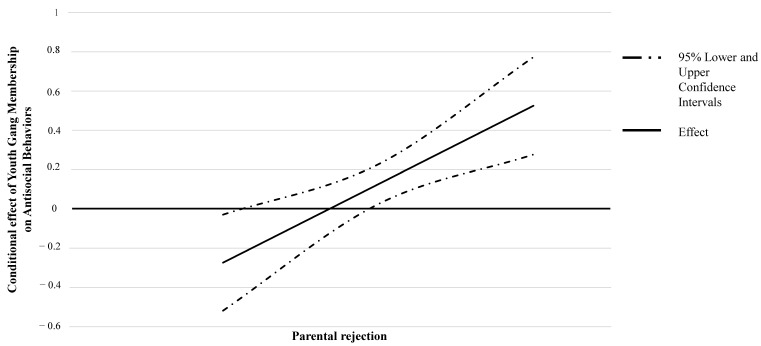
The effect of Youth Gang Membership on Antisocial Behaviors conditional on levels of Parental Rejection.

**Table 1 ijerph-17-08791-t001:** Descriptive statistics (Means and SDs) and ANOVA’s results for the three groups (deviant gang members vs. non-deviant gang members vs. non-gang members) on involvement in Antisocial Behaviors (ASBs).

	Deviant Gang Members (*n* = 157)		Non-Deviant Gang Members (*n* = 168)		Non-Gang Members (*n* = 492)			
Measure	M	SD	M	SD	M	SD	F	Ƞ2
Antisocial Behaviors (ASBs)	1.81 a	0.76	1.35 b	0.49	1.42 b	0.66	25.70 ***	0.06

Note: Univariate Post-hoc Analysis. Means with different superscripts letters ^(a, b)^ indicate statistically significant differences between the deviant gang members group and the other two groups, as determined by Bonferroni′s multiple-comparison test. *** *p* < 0.001.

**Table 2 ijerph-17-08791-t002:** Correlations among study’s variables, means (M) and standard deviations (SDs).

	1	2	3	4	5	6	7	M	SDs
1. Self-Serving Cognitive Distortions	1							2.28	1.01
2. Parental Acceptance-Rejection	0.364 *	1						1.65	0.49
3. Community Violence Witnessing	0.330 *	0.141 *	1					1.80	0.88
4. Youth Gang Membership	0.325	0.104	0.235 *	1				-	-
5. Antisocial Behaviors	0.506 *	0.528 *	0.368	0.240	1			1.48	0.67
6. Gender	−0.128 *	−0.052	−0.189 *	−0.139 *	−0. 165 *	1		-	-
7. School Grade	0.225 *	0.162 *	0.042	−0.073	0.194 *	0.182 *	1	-	-

Note: Youth Gang Membership was coded 0 for non-gang members, 1 for gang members. Gender was coded 1 for males, 2 for females. School Grade was coded 1 for middle, 2 for high school students. * = significant results for *p* < 0.002 (*p*-value of 0.05 adjusted using Bonferroni correction for multiple comparisons).

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
