# Peer review of "Individual, Familial, and Socio-Environmental Risk Factors of Gang Membership in a Community Sample of Adolescents in Southern Italy"

_ijerph, 2020, doi:10.3390/ijerph17238791_

Round 1

Reviewer 1 Report

We thank the reviewer for his/her helpful suggestions and constructive feedback on our original submission. After addressing the issues raised, we feel the quality of the paper is much improved and hope the reviewer will agree. We reply to each comment in a point-by-point fashion, with our responses in red italics.

Measures

Cronbach's alpha needs to be calculated for all subscales in the questionnaire PARQ_child (line 313)

We thank the reviewer for raising this point. Cronbach’s alpha values for each subscale of the PARQ were:

Mother subscales

Warmth: .86; hostility: .86; neglect: .77; rejection: .84.

Father subscales

Warmth: .87; hostility: .87; neglect: .76; rejection: .83.

For the sake of simplicity, we have now reported in the text the overall range of values (Line 349-350).

Follow the same pattern and use the acronym of the Exposure to Community Violence Questionnaire (line 328)

We thank the reviewer for this suggestion. The acronym ECVQ has been now used throughout the scale description.

Follow the same pattern to describe the scales (line 340)

We thank the reviewer for this suggestion.

In the adjustment indexes you must report the TLI (line 349)

The TLI index was .92. It has been now reported in the text.

Data Analysis (line 355)

Justify the parametric data 

In the data analysis section, we have now reported the investigation of univariate normality of data distribution, revealing that no study’s variables approached skewness>|3| or kurtosis >|10|. We hope the reviewer will be satisfied with this response. LINE 394-395.  

It is important to specify the particular analyses you have done of anova, as well as references to the effect size

We thank the reviewer for raising this point. We have now reported in this section the specific ANOVA test we performed, including the Bonferroni post hoc tests that we used to analyze the pairwise associations. We have now specified also that the significance level was set at p<.05, and that we used eta squared to interpret levels of effect size according to Cohen's guidelines (1988). (line 395-400).

Results

(Bonferroni Post-hoc Test): The differences marked in the Table, to make it easier to interpret (line 409)

We have now included in the table caption a more detailed explanation of how interpreting significant differences between groups.

The first best sentence in data analysis (line 422), as well as the codifications.

We are not sure to have correctly interpreted this reviewer’s comment. We have now better clarified the interpretation of correlations, especially when referring to categorial variables.

Report correlation statistics r= .333 (with 3 decimals) (line 432)

In reporting the results in the original submission, we referred to the APA style guidelines (6th edition), according to which it’s suggested to report two decimals. We have now reported 3 decimals, as recommended.

Reviewer 2 Report

The authors are to be congratulateds for their focus on research that provides a significant (rather than a ‘novel’) contribution to gang literature, particularly with regards to idiosyncratic factors related individual, familial and soci-environmental factors within the Italian context.  As the authors point out, the research/paper moves in the direction of contributing to the gap of YGM in an Italian community and risk factors on involvement in antisocial behaviors. My comments are included in the section below:

Individual, familial and socio-environmental risk factors of gang membership

The authors are to be congratulated for their focus on research that provides a significant (rather than a ‘novel’) contribution to gang literature, particularly with regards to idiosyncratic factors related individual, familial and soci-environmental factors within the Italian context.  As the authors point out, the research/paper moves in the direction of contributing to the gap of YGM in an Italian community and risk factors on involvement in antisocial behaviors.

We thank the reviewer for his/her helpful suggestions and constructive feedback on our original submission. After addressing the issues raised, we feel the quality of the paper is much improved and hope the reviewer will agree. We reply to each comment in a point-by-point fashion, with our responses in red italics.

We thank the reviewer for his/her appreciation on our research. We have received many of his/her suggestions and reorganized the paper according to them.

Abstract

The abstract helps to outline the details of the study, however the authors may like to consider whether there is a need to add so much quantitative detail of the findings of the study in the abstract.  There are also numerous grammatical errors in the abstract that impacts on flow and readability.  It would also be of value for the first sentence of the abstract to be used to set the scene perhaps of % of YGM in Italy or a sentence that sets the scene and problem identified that motivated the focus of the study.  Perhaps similar wording to the wording used at the bottom of page 5 of your paper, “Despite the growing social alarm generated in Italy by the recurrent news concerning violent episodes involving youth gangs, systematic research in this field, especially within a psychological framework, is still limited”.

We thank the reviewer for noticing some critical issues in the abstract. We have now added more details and rephrased some sentences based on the provided suggestions.

Intro section

It is great that the authors set the scene for the public health concern, in the US and Europe, however, it would be of value to see if, in this first paragraph more current statics, particularly for Europe can see sourced and included to provide the reader with an appreciation of ‘currency of the issue’.    Is there the potential perhaps to include some similar wording to that used on page 6 “According to the most recent official data from the Public Security Department of the Italian Ministry of the Interior (2018), the number of reported crimes in Italy is approximately 6,600 per day. The metropolitan area of Naples is the second in Italy for reported crimes, with 568.91 for every 100,000 inhabitants.”

We specified that we do not have comparative data for Europe. In order to reduce the length of the introduction, we reduced the space we originally dedicated to US data.

It is also important to clearly define and provide a definition up front early in the paper of how the authors define YGM.  Perhaps the definition is outlined on Line 41-44 of page 1, but the reader would value from a definition being clearly stated.

We thank the reviewer for this suggestion. We moved the critical issues concerning the definition of YGM at the beginning of the paragraph (line 41-46).

Page 2 Line 58-59 – It is unclear what is meant by the following : “it is not obvious that the US gangs’ features would overlap with the gangs’ features in other countries. In this regard, some authors refer to the “gang paradox” concerning a stereotyped 

We agree that this sentence is not clear. We deleted the reference to the gang paradox clearly explaining that features of American gangs are not applicable to European and Italian gangs. (line 75)

Page 3 Line 100-101 – Meaning unclear – “This percentage exactly corresponded to the average involvement in the whole international sample.

We have now specified that the percentage of Italian gang members in this research was similar to the mean percentage across countries. (line 121)

Page 3 – 103-108 – One long sentence…needs breaking up.

We thank the reviewer for this suggestion.

The wording below requires breaking into several sentences…

Page 4 - Overall, although the findings discussed above seem to confirm the facilitating role of moral neutralization mechanisms in YGM, to our knowledge, no study

Line 151 - A great number of studies ?have focused

We thank the reviewer for this suggestion. We shortened all long sentences, as recommended.

You note on page 4 Line 162 that your study focusses on “the quality of the parent–child relationship” ….therefore it is important to define for your study how this is determined and defined.

We agree that the emphasis on the quality of parent—child relationship is confounding. We have now directly introduced the topic of parental acceptance. (line 184-185)

The literature review for this paper is interesting and extensive (nearly 5 pages of the paper). However, I am concerned that more attention is given to the background literature than to the discussion (only 3 pages) implications (0) and conclusion (.5).  I would recommend that the authors tighten up the literature review to the key themes of the paper and add an additional page to the discussion and implications section.  In this way they are able to provide a significant contribution to the field regarding this issue.

We thank the reviewer for this suggestion. We have now deleted some paragraphs to let the reader stay focused on the key themes of the paper. We hope the reviewer will appreciate our effort in doing that.

Method – Participants – I am confused re mention at the beginning of this section to the following: “The study design involved eighth and eleventh graders of two middle and three high schools in 269 Arzano…” Yet later in this section you mention: “Participants were from 10 different schools located in the 280 metropolitan area of Naples, with most living in Arzano (71.4%)”  Therefore there seems to be a discrepancy in the number of schools mentioned here.

Thank you for noting this typo. The schools involved in the study were five. We have now corrected this information.

Method and analysis were thorough and sound, well done.  I do wonder whether it would have been of value to analyse the findings of this study re ‘factors’ through the use of a social ecological framework.  This point is also raised within the discussion section below.

Discussion

The authors are encouraged to more fully explore details on the ‘social-ecological approach’ mentioned on page 13 of the paper.  There is significant literature and quite a number of SE frameworks that could have been used to make sense of their findings (See several examples of excellent papers and books outlined at the end of this review).  This is where I think the paper unfortunately falls down.  At this point the mention simply to Bronfebrenner appears tokenistic to the mention of any effort to make sense of the research findings in relation to a social ecological frame.  The impact is that there are numerous mentions within the discussion of areas where further study could be conducted to explore some of these factors.

Further, there it is important to more comprehensively expand on  other risk or protective factors that would help modify or ‘neutralize’ a deviant behaviour.  However, I do acknowledge wording such as the following that is integrated into this section “it is reasonable to hypothesize  that pre-existing cognitive mechanisms, such as the tendency to minimize the consequences of deviant behavior as well as the tendency to perceive the world as a hostile, threatening, or dangerous place, find fertile ground in gang affiliation.” 

I would suggest that if the authors were able to refer to a Figure or Image of their findings and associated factors through the use of a SE framework, this would not only help the reader to appreciate the complexity of this phenomenon and associated factors, but would help to support statements such as ‘it is reasonable to hypothesize”, where they could refer to the framework to show the perverseness of these types of factors as well as other variables and potential factors at play within this contextual setting.

Furthermore, in the present study the authors write, “Our findings add a novel contribution to the gang literature focusing on a specific component of the general concept of 5neighborhood disorganization, supporting the idea that living and being exposed to the community violence are intrinsically linked to gang membership. This link can be explained by several theoretical models…..”  Howevever, if the authors initially refer to a SE framework and could potentially point to or refer to their findings displayed through a SE framework they would be able to then further make sense of the factor above, to compliment reference to SL theory and models of behavior.

We really appreciate this reviewer’s suggestion. We carefully read the suggested papers and have now integrated other details of the socio-ecological approach in the discussion section (line 605-611). We specifically focused on the neighborhood level, stressing the importance of the social capital and the moral capital roles. We hope the reviewer will appreciate our effort.

It is suggested that the authors hesitate in writing the following, “These findings should be generalized with caution because the family role could be highly context related.”  Rather, if the authors refer to key factors from their study identified using a SE framework they could point out that the findings are highly contextualised and so would not encourage ‘generalising’, but perhaps what could be learnt from this study is the multiple contextual factors at play the impact on the translation of these findings being difficult to other settings.

We totally agree with this reviewer’s comment. We have now clarified these aspects of the results generalization in the conclusions (line 696-702), highlighting that several other systems of the individual’s ecology should be aknowledged, and that the complexity of the human-environment transaction requires a multidisciplinary approach, considering the different levels of analysis of the phenomenon (psychological, social, legal, criminological).

This supports the authors wording below:

****”it is important to expand research throughout multiple geographical and socio-cultural areas to investigate characteristics of juvenile gangs.”  But then consider commenting on how this could happen….and why this is important.

We really appreciate all the reviewer’s suggestions and his/her support in wording the required conceptualizations.

Grammatical tightening required in abstract and throughout paper

We thank the reviewer for raising this point. We have carefully checked the document and made the appropriate linguistic revisions.

….. Furthermore, we examined the mediating and/or moderating role of 15 YGM in (of?) the association between risk factors and involvement in antisocial behaviors (ASBs).

As we referred to the (mediating and moderating) “role”, we preferred to use “in the association”, as it is frequently reported in the scientific literature. We hope the reviewer will agree with this choice.

****Again here… These findings pointed out the role of distorted moral cognitions and 24 experience of violence witnessing within community as fertile ground for gang involvement 25 highlighting the need to consider both individual and contextual factors in preventing and reducing 26 adolescents’ risk of joining a gang.

We thank the reviewer for noticing this point. We have now broken this sentence to enhance its readability.

***Note below that there are quite a number of these types of errors that require a thorough read and grammatical updating to ensure flow and readability.

We thank the reviewer for raising this point. We have carefully checked the document and made the appropriate linguistic and stylistic revisions.

Page 2 – line 74-75 - thus contributing to the debate around the still open question whether YGM enhances ASBs or whether the role of a gang is limited to previously 75 antisocial individuals.

We thank the reviewer for noticing this point. We have now rephrased the sentence: “… or whether only previously antisocial individuals deliberately choose to become gang members”.

in a study performed with a large community sample of over 5,000 adolescents found that 17% of 94 participants self-evaluated belonging to a gang. In this study, they found that about one third of gang

We thank the reviewer for noticing this point. We have now rephrased the sentence (line 112-115).

Page 3 – 130-131***Multiple sentences need the addition of a comma to help provide clarity and readability…. - euphemistic labeling, displacement of responsibility, and blaming others were the ones? (clarify ‘ones) more 131 frequently used by gang members than their non-gang counterparts.

We thank the reviewer for noticing this point. We have now clarified that “ones” refers to the mechanisms more frequently used by gang members than their non-gang counterparts (line 150-153)

A few great references and sources re social ecological frameworks and approaches

Lounsbury, D., & Mitchell, S. (2009). Introduction to special issue on social ecological approaches to community health research and action. American Journal of Community Psychology, 44(3-4), 213-220.

Stokols, D. (2018). Social ecology in the digital age: Solving problems in a globalised world. San Diego, California: Academic Press.

Stokols, D., Lejano, R., & Hipp, J. (2013). Enhancing the resilience of human-environment systems:  A social ecological perspective. Ecology and Society 18(1).

We thank the reviewer for these valuable suggestions.

Reviewer 3 Report

The submitted manuscript intends to shed light into an important social problem in south communities in Italy: the so called Mafia. Authors made a big effort to systematically observe the impact of different individual and social variables that may predict the approach of adolescents to these criminal associations. They include a considerable sample of adolescents (817 adolescents) identified at high risk. All in all, although the present paper provides a valuable research because of its preventive approach to a real social problem, some aspects might be addressed.

We thank the reviewer for his/her helpful suggestions and constructive feedback on our original submission. After addressing the issues raised, we feel the quality of the paper is much improved and hope the reviewer will agree. We reply to each comment in a point-by-point fashion, with our responses in red italics.

We thank the reviewer for this appreciation on our study.

  1. Authors should provide a more detailed description of the socioeconomic characteristics of adolescents’ environment.

We have now added a more detailed description of the socio-economic characteristics of the sample.

     2. For the correlations, multiple comparisons correction should be performed.

We thank the reviewer for this helpful suggestion. We have now applied the correction for multiple comparisons for correlations to support the interpretation of the findings, as recommended.

    3. One clear weakness of this study is that it relays exclusively in self-report measures, with adolescents as the only informants. No other complementary objective measures were collected. This limits the extent to which the results might be interpreted. Authors should discuss also this limitation.

We thank the reviewer for this suggestion. We totally agree that it would be important to have information from other informants or objective measures such as police reports. We have now discussed this limitation in the paper (line 662-670).

Round 2

Reviewer 3 Report

In this second version of the manuscript authors made a great effort to improve it following reviewers’ recommendations. I thank authors for the job. To my view, authors addressed reviewers' comments satisfactorily. I just recommend authors to pay attention to some format (e.g. table 2) and English style minor issues in a careful re-editing.

We are pleased to have the opportunity to submit a revise version of our paper entitled “Individual, Familial, and Socio-Environmental Risk Factors of Gang Membership in a Community Sample of Adolescents in Southern Italy” (ijerph-982631) for publication in the International Journal of Environmental Research and Public Health.

In this revised version of the manuscript we have carefully made the appropriate linguistic revisions, checked some aspects related to some format (e.g. table 2) and added the project identification code (file reference number 3/2020, dated 13/01/2020) within the manuscript.

Author Response

Dear Editor,

We are pleased to have the opportunity to submit a revise version of our paper entitled “Individual, Familial, and Socio-Environmental Risk Factors of Gang Membership in a Community Sample of Adolescents in Southern Italy” (ijerph-982631) for publication in the International Journal of Environmental Research and Public Health.

In this revised version of the manuscript we have carefully made the appropriate linguistic revisions, checked some aspects related to some format (e.g. table 2) and added the project identification code (file reference number 3/2020, dated 13/01/2020) within the manuscript.

After addressing the issues raised, we feel the quality of the paper is much improved and hope you agree.